# Cranial Structure of *Varanus komodoensis* as Revealed by Computed-Tomographic Imaging

**DOI:** 10.3390/ani11041078

**Published:** 2021-04-09

**Authors:** Sara Pérez, Mario Encinoso, Juan Alberto Corbera, Manuel Morales, Alberto Arencibia, Eligia González-Rodríguez, Soraya Déniz, Carlos Melián, Alejandro Suárez-Bonnet, José Raduan Jaber

**Affiliations:** 1Instituto Universitario de Investigaciones Biomedicas y Sanitarias (IUIBS), Facultad de Veterinaria, Universidad de Las Palmas de Gran Canaria, Trasmontaña, Arucas, 35413 Las Palmas, Spain; saraperezalberto@gmail.com (S.P.); juan.corbera@ulpgc.es (J.A.C.); manuel.morales@ulpgc.es (M.M.); eligia.gonzalez102@alu.ulpgc.es (E.G.-R.); 2Hospital Clínico Veterinario, Facultad de Veterinaria, Universidad de Las Palmas de Gran Canaria, Trasmontaña, Arucas, 35413 Las Palmas, Spain; mencinoso@gmail.com (M.E.); soraya.deniz@ulpgc.es (S.D.); carlos.melian@ulpgc.es (C.M.); 3Departamento de Morfologia, Facultad de Veterinaria, Universidad de Las Palmas de Gran Canaria, Trasmontaña, Arucas, 35413 Las Palmas, Spain; alberto.arencibia@ulpgc.es (A.A.); alejandro.suarez@ulpgc.es (A.S.-B.)

**Keywords:** computed tomography, head, Komodo dragon

## Abstract

**Simple Summary:**

We investigated the head of Komodo dragons using CT imaging. Cross-sections show that all cranial bones can be delineated, while soft tissue structures are evident but not clearly identifiable without an anatomical atlas. Additional three-dimensional reconstructed and maximum intensity projection images of the head were presented to depict bony structures. The anatomical structures identified on the CT images could help further assess the head of the Komodo dragon.

**Abstract:**

This study aimed to describe the anatomic features of the normal head of the Komodo dragon (*Varanus komodoensis*) identified by computed tomography. CT images were obtained in two dragons using a helical CT scanner. All sections were displayed with a bone and soft tissue windows setting. Head reconstructed, and maximum intensity projection images were obtained to enhance bony structures. After CT imaging, the images were compared with other studies and reptile anatomy textbooks to facilitate the interpretation of the CT images. Anatomic details of the head of the Komodo dragon were identified according to the CT density characteristics of the different organic tissues. This information is intended to be a useful initial anatomic reference in interpreting clinical CT imaging studies of the head and associated structures in live Komodo dragons.

## 1. Introduction

The introduction of imaging diagnostic techniques has revolutionized the knowledge in reptile medicine. The radiographic evaluation has been traditionally used by clinicians [1]. Nonetheless, the progressive increase in modern imaging modalities such as computed tomography (CT) and magnetic resonance imaging has improved diagnostic abilities in reptile medicine and research [2]. Therefore, these techniques represent an enormous resource that allows for fast, non-invasive anatomy visualizations of internal structures that are challenging to interpret [2].

In recent years, the contributions of zoo veterinarians, researchers, and specialized technicians (anatomists, radiologists, and wildlife and exotic specialists) working with captive and free-ranging animals to prevent and treat diseases that threaten the survival of species in wildlife conservation have increased [3]. Since 1996, the Komodo dragon (*Varanus komodoensis*) is listed as vulnerable by the Red List of the World Conservation Union [4]. To our knowledge, the anatomy of different species of reptiles has already been thoroughly described by radiology and CT [1,5,6,7,8,9,10], but only sparse numbers of these studies reported comprehensive descriptions of computed tomographic features of the head [1,5,6,9,10,11,12]. To date, not one of these reports investigates to what extent structures of the varanid head could be visualized and identified in low-resolution clinical CT-image data. In the Komodo dragon and other reptiles, the head conforms to a complex structure, which is challenging to interpret. The purpose of this study was to describe the normal anatomy of the head of the *Varanus komodoensis* by computed tomography, and three-dimensional head reconstructed images to assist in the understanding of the head and its associated structures.

## 2. Materials and Methods

### 2.1. Animals

Two 17-year-old female specimens born in captivity at Reptilandia Park (Las Palmas, Spain) were imaged at the Veterinary Clinic Hospital of Las Palmas de Gran Canaria University. One female had a length of 225 cm (snout-vent length) and weighed 36 kg, whereas the other had a length of 190 cm and weighed 24 kg. No physical abnormalities were detected before the study. The Ethical Committee of Las Palmas de Gran Canaria University, College of Veterinary Medicine Section authorized the research protocol (MV‒2019/04). The owner of the animals was informed of the study and signed consent for participation in the study.

### 2.2. CT Technique

Sequential transverse CT slices were obtained using a 16‒slice helical CT scanner (Toshiba Astelion, Toshiba Medical System, Madrid, Spain). The animals were positioned symmetrically in ventral recumbency on the CT couch, and a standard clinical protocol (120 kVp, 80 mA, 512 × 512 acquisition matrix, 1809 × 858 field of view, a spiral pitch factor of 0.94, and a gantry rotation of 1.5 s) was used to acquire sequential transverse CT images of 1 mm thickness slice. The original transverse data were stored and transferred to the CT workstation. No CT density or anatomic variations were detected in the head of the dragons used in the study. In the CT technique, tissue density can be assessed directly on the image using the Hounsfield Unit scale, The range of values assumes 2000 shades of gray (between −1000 and +1000 HU). However, as the human eye cannot distinguish more than 30 shades, representing the entire range of values in an image implies not being able to visualize a large amount of information. Therefore, only a partial sector of the TC values previously selected by the operator (window selection) is represented by grayscale. Ultimately, the use of windows allows extracting the information that the computer has, showing only a part of it, which is of interest in each anatomical region. Therefore, bone window, soft tissue windows, brain window, and pulmonary window can be applied, delivering alternate streams of information. Thus, two CT windows were applied by adjusting the window widths (WW) and window levels (WL): a bone window setting (WW = 1500; WL = 300), and a soft tissue window setting (WW = 350; WL = 40). The original data were used to generate head volume-rendered reconstructed images after manual editing of the transverse CT images to remove soft tissues using a standard dicom 3D format (OsiriX MD, Geneva, Switzerland). In addition, maximum intensity projection (MIP) images were obtained to better display the outlines between bones and other lower-attenuation structures using an image viewer (OsiriX MD, Apple, Cupertino, CA, USA). MIP is a specific type of rendering in which the brightest voxel is projected into the 3D image. There tends to be much less variability in MIP image reconstruction than in volume rendering because fewer parameters are factored into the MIP algorithm [13].

## 3. Results

### 3.1. Transverse Computed Tomography Images

Transverse sections are provided that demonstrate critical anatomical features of the varanid cranium (Figure 1, Figure 2, Figure 3, Figure 4, Figure 5 and Figure 6). Figure 1 consists of three images: (C) Sagittal image of the head, where each line and number (I–VI) represents the approximate level of the following transverse CT images, (A) CT bone window, (B) CT soft tissue window. Figure 2, Figure 3, Figure 4, Figure 5 and Figure 6 represent transverse CT images where (A) CT bone window, and (B) CT soft tissue window. The CT images are presented in a cranial to caudal progression from the septomaxilla level (Figure 1) to the brain stem level (Figure 6). The comparison between available literature and CT images enabled us to identify most of the clinically relevant anatomic structures of the head. These features were identified according to location and the degree of attenuation of the different tissues.

With regards to hard tissues, the CT images acquired using the bone window setting (Figure 1A, Figure 2A, Figure 3A, Figure 4A, Figure 5A, Figure 6A) provided good differentiation between the bones and the soft tissues of the head. Thus, the bones of the cranium (prefrontal, frontal, postorbital, parietal, squamosal, quadrate, jugal, pterygoid, basioccipital, parabasisphenoid, and maxilla), the mandible (dentary, angular, surangular, and articular bones) and hyoid bones were easily recognizable because of the high CT density in cortical bone and the low CT density in their medullary cavities. Most of these structures were also visualized with the soft tissue window setting (Figure 1B, Figure 2B, Figure 3B, Figure 4B, Figure 5B, Figure 6B).

Air-filled structures, such as the nasal cavity, larynx, trachea, and the oral cavity gave negligible CT-tissue density and appeared black with both window settings.

Soft-tissue structures—such as the jaw muscles, the labial and nasal glands, the eyes, and the Harderian glands—gave an intermediate CT density and appeared grey. The nervous structures (brain, cerebellum, lateral ventricles, brain stem, and spinal cord) were appreciated in both CT window modalities (Figure 3, Figure 4, Figure 5 and Figure 6).

### 3.2. Head Volume-Rendered Reconstructed Images

We provide images of the three-dimensional structure of the Varanus cranium in dorsal and ventral view (Figure 7 and Figure 8, respectively) and the left lateral view (Figure 9). Volume-rendered reconstructed CT images provided good visualization of the different bones that compose the skull. Thus, the orbital border was circumscribed by the lacrimal, the prefrontal, and the jugal bones (Figure 7 and Figure 9). Moreover, the jugal bone was distinguishable from the ectopterygoid (Figure 9). At the posterodorsal border of the orbit, the fusion of postorbital and postfrontal bones could be seen in the lateral and dorsal reconstructed CT images (Figure 7 and Figure 9). In ventral view, the following bones of the neurocranium were clearly delineated: the parabasisphenoid, the basioccipital, and the prootic (Figure 8). The junction between premaxilla and maxilla with the tooth arranged in a curved row was identified in the lateral and ventral view (Figure 8 and Figure 9). In the lateral view, this tooth row curved with the margin of the mandible and maxillary. Besides, the primary curvature of the maxilla was convex, whereas that of the mandible was concave. In addition, the coronoid process was quite prominent, and the surangular and articular bones were observed extending caudally (Figure 9).

### 3.3. Maximum Intensity Projection (MIP) Images

Two MIP images corresponding to dorsal (Figure 10) and ventral (Figure 11) views of the varanid skull were selected. These images were able to resolve the relation between the bones that comprise the head. The dorsal MIP image showed the junction between the premaxilla and the maxilla. We were also able to show how the laminar disposition of the vomer supports the septomaxilla (Figure 10). This last finding could be better distinguished in the ventral MIP image (Figure 11). The relation between the lacrimal, the prefrontal, and the frontal bones was seen in the dorsal view (Figure 10). At the posterodorsal border of the orbit, the fusion of postorbital and postfrontal bones could be easily seen. In addition to these observations, the junction of the frontal and the parietal bones were identified in dorsal (Figure 10) and ventral (Figure 11) MIP images. The ventral MIP image showed excellent visualization of the pterygoid, a flat, Y-shaped bone. This bone provides a rounded process that contacts the caudal border of the palatine. Moreover, this view displayed the junction between the parabasisphenoid, the prootic, and the basioccipital bones. This last bone forms the ventral portion of the occipital condyle.

## 4. Discussion

In recent years, the contribution of imaging techniques to reptile medicine has increased the knowledge in veterinary practice and research [4,5,6,12]. Traditionally, radiography and ultrasonography have been used to obtain information on the bony and the main soft-tissue structures of different reptile regions [14,15]. More advanced imaging techniques, such as computed tomography, have become increasingly common in veterinary clinical practice [15]. Third and fourth generation CT scanners give considerable advantages over traditional radiography: body sections from different tomographic planes, fair anatomic resolution without superimposition of the tissues, and a higher differentiation of tissue densities allow better detection of several diseases [6,15,16]. Nonetheless, its use in reptile medicine is still limited because of the cost of the equipment, availability, and logistic problems of acquiring CT images in these animals [6,15].

The cranium of the genus *Varanus* is a complex structure that has received some attention in morphofunctional studies [12,17], perhaps due to the enormous disparity in the form that evolved among varanid lizards [18]. The head of this iconic varanid represents a complex structure, composed of various tissues with varying degrees of attenuation in radiographic images, making it a challenging object to assess. The two window settings used in our CT study facilitated the identification of the main head structures such as the bones of the skull, mandible, muscles, air-filled structures of the respiratory and digestive system, the nervousranium and other associated structures. Visualizing images through use of the “bone window” provided good resolution for skeletal structures, whereas the “soft tissue window” allowed us to distinguish the eyes and the nervous structures from the remaining soft tissues. Similar results were described in other studies conducted in reptiles [1,5,6,15]. Several causes have been reported to explain the low resolution of soft tissue structures showed in our study [1,15], such as the small volume of these species, the impossibility of reducing the field of view of the CT scanner, and the presence of bones embedded within the skin. These bones, called cephalic osteoderms, vary in shape and complexity and serve primarily as a defensive anatomical system to protect individuals during aggressive confrontations with other specimens [11]. To avoid this low resolution, some investigations reported the use of micro-CT scanners [17], although this equipment is not usually available in veterinary clinics [15].

Employing computed tomography, we were able to fully visualize the cranium in virtual reconstructions and MIP images. Thus, the reconstructed images showed a broad, dorsoventrally compressed cranium. The mandible was curved, and teeth were laterally compressed. This morphology contrasts with most other varanids, which feature a relatively narrow rostrum, a dorsoventrally tall cranium, and a straight ventral margin of the maxilla [18]. Additionally, an enormous variation of the orbit was observed, especially along the posterior margin of the orbit, which is closed in lizards, or semiclosed in these varanids. This fact is determined by variation in the shape, size, and presence of the jugal bone and variations in the postorbital and postfrontal bones [19]. MIP images proved a helpful tool in delineating bones in volume-rendered images.

In conclusion, the CT images obtained in this study provided an adequate anatomical interpretation of the head of *Varanus komodoensis*. This information could be used to diagnose disorders involving the head of lizards, such as abscesses, metabolic bone diseases, fractures, and neoplasia.

## Figures and Tables

**Figure 1 animals-11-01078-f001:**
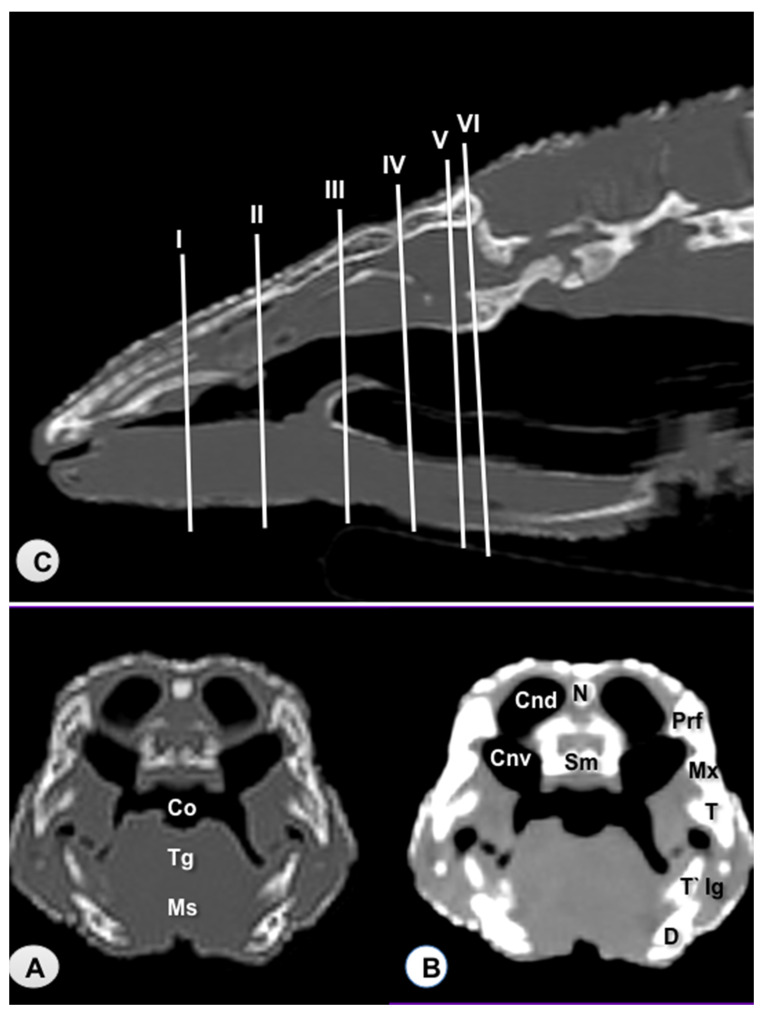
Sagittal image of the head of *Varanus komodoensis*. The lines and numbers (I–VI) represent the approximate level of the following transverse CT images (**C**). Transverse CT image of the head at the level of the nasal cavity corresponding to line I. (**A**) CT bone window. (**B**) CT soft tissue window. These images are displayed so that the right side of the head is to the viewer’s left and the dorsal view is at the top. N: Nasal bone. Sm: Septomaxilla. Cnd: Dorsal nasal conchae. Cnv: Ventral nasal conchae. Prf: Prefrontal bone. 6. Maxillary bone. T: Tooth. T’: Tooth. Co: Cavum oris. Tg: Tongue. Ms: Musculus intermandibularis + Musculus geniohyoideus + Musculus genioglossus. D: Dentary bone. Ig: Infralabial glands.

**Figure 2 animals-11-01078-f002:**
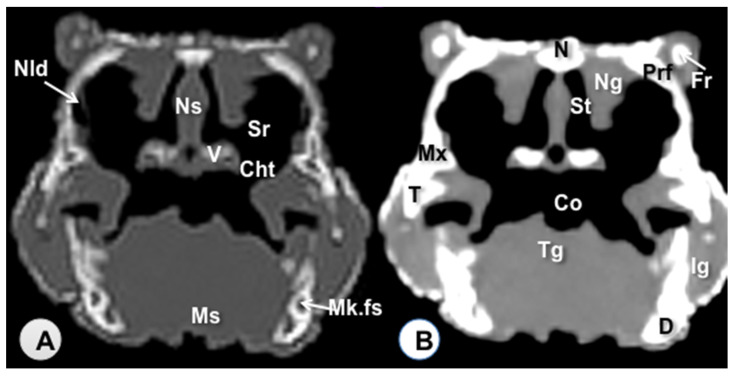
Transverse CT image of the head of *Varanus komodoensis* at the level of the nasal bone corresponding to line II. (**A**) CT bone window. (**B**) CT soft tissue window. These images are displayed so that the right side of the head is to the viewer’s left and the dorsal view is at the top. N: Nasal bone. Prf: Prefrontal bone. Ns: Nasal septum. Ng: Nasal glands. St: Stammteil, V: Vomer. Sr: Subconchal recess. Cht: Choanal tube. Nld: Nasolacrimal duct. Mx: Maxillary bone. T: Tooth. Co: Cavum oris. Tg: Tongue. Ms: Musculus intermandibularis + Musculus geniohyoideus + Musculus pterygoideus + Musculus hyoglossus. D: Dentary bone. Mk.fs: Meckelian fossa. Ig: Infralabial glands. Fr: Frontal bone.

**Figure 3 animals-11-01078-f003:**
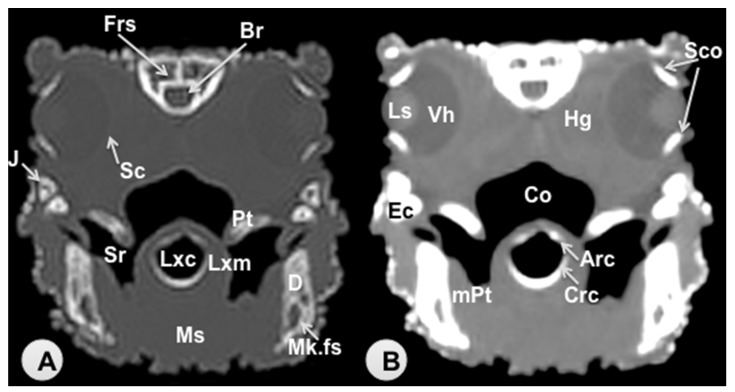
Transverse CT image of the head of *Varanus komodoensis* at the level of the eyes corresponding to line III. (**A**) CT bone window. (**B**) CT soft tissue window. These images are displayed so that the right side of the head is to the viewer’s left and the dorsal view is at the top. Frs: Frontal sinus. Br: Brain. Sco: Scleral ossicles. Ls: Lens. Vh: Vitreous humor. Hg: Harderian gland. Sc: Sclera. J: Jugal bone. Ec: Ectopterygoid bone. Pt: Pterygoid bone. D: Dentary bone. Mk.fs: Meckelian fossa. mPt: Musculus pterygoideus. Lxc: Laryngeal cavity. Lxm: Laryngeal muscles. Ms: Musculus intermandibularis + Musculus geniohyoideus + Musculus genioglossus + Musculus hyoglossus. Crc: Cricoid cartilage. Arc: Arytenoid cartilage. Co: Cavum oris. Sr: Sublingual recess.

**Figure 4 animals-11-01078-f004:**
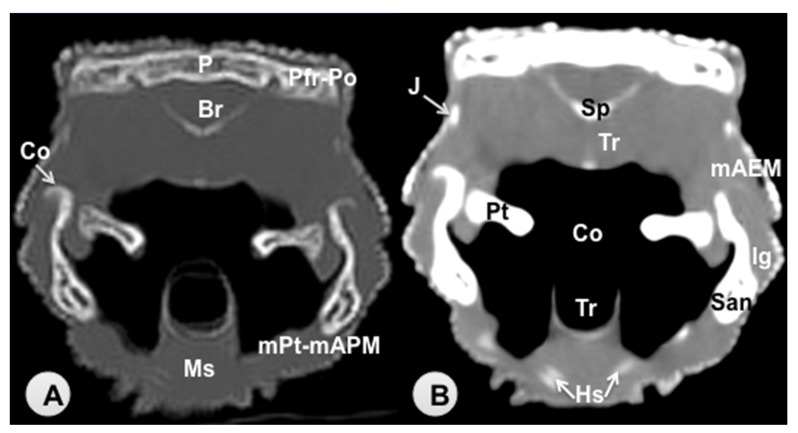
Transverse CT image of the head of *Varanus komodoensis* at the level of the parietal bone corresponding to line IV. (**A**) CT bone window. (**B**) CT soft tissue window. These images are displayed so that the right side of the head is to the viewer’s left and the dorsal view is at the top. P: Parietal bone (frontoparietal suture). Pfr-Po: Postfrontal + Postorbital bone. Br: Brain. Sp: Sphenoid bone. Pt: Pterygoid bone. San: Surangular bone. Co: Coronoid bone. Co: Cavum oris. Tr: Trachea. Hs: Hyobranchial skeleton. Ms: Musculus intermandibularis + Musculus geniohyoideus + Musculus hyoglossus. J: Jugal bone. Ig: Infralabial glands. mAEM: Musculus adductor mandibulae externus. mPt-mAPM: Musculus pterygoideus + Musculus adductor mandibularis posterior.

**Figure 5 animals-11-01078-f005:**
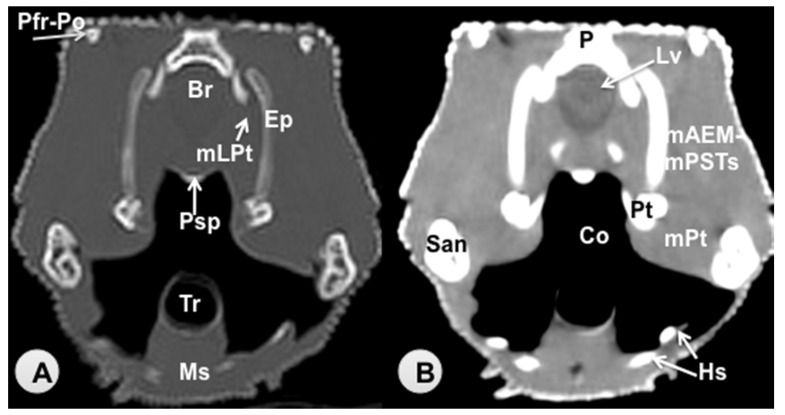
Transverse CT image of the head of *Varanus komodoensis* at the level of the postorbital + postfrontal bone corresponding to line V. (**A**) CT bone window. (**B**) CT soft tissue window. These images are displayed so that the right side of the head is to the viewer’s left and the dorsal view is at the top. P: Parietal bone. Lv: Lateral ventricle. Br: Brain. Ep: Epipterygoid bone. Pt: Pterygoid bone. Psp: Parabasisphenoid bone. Pfr-Po: postfrontal + postorbital bone. San: Surangular bone. Co: Cavum oris. Tr: Trachea. Hs: Hyobranchial skeleton. Ms: Musculus intermandibularis + Musculus geniohyoideus + Musculus hyoglossus. mAEM-mPSTs: Musculus adductor mandibulae externus + Musculus pseudotemporalis superficialis. mLPt: Musculus levator pterygoideus. mPt: Musculus pterygoideus.

**Figure 6 animals-11-01078-f006:**
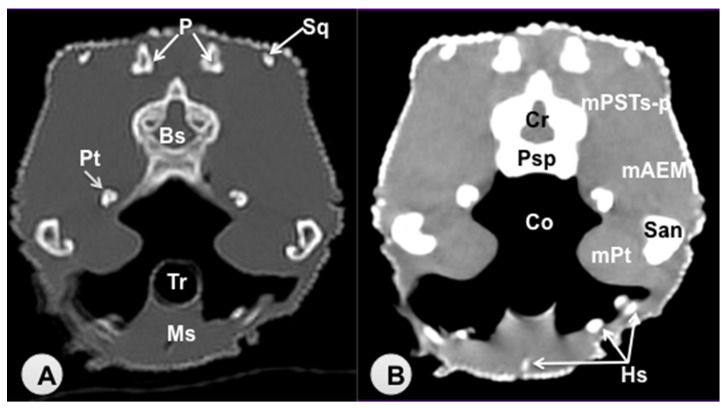
Transverse CT image of the head of *Varanus komodoensis* at the level of the squamosal bone corrsponding to line VI. (**A**) CT bone window. (**B**) CT soft tissue window. These images are displayed so that the right side of the head is to the viewer’s left and the dorsal view is at the top. Sq: Squamosal bone. P: Parietal bone. Cr: Cerebellum (vermis). Bs: Brain stem. Psp: Parabasisphenoid bone. Pt: Pterygoid bone. San: Surangular bone. Co: Cavum oris. Tr: Trachea. Hs: Hyobranchial skeleton. Ms: Musculus intermandibularis + Musculus geniohyoideus + Musculus hyoglossus. mAEM: Musculus adductor externus mandibularis. mPSTs-p: Musculus pseudotemporalis superficialis and profundus. mPt: Musculus pterygoideus.

**Figure 7 animals-11-01078-f007:**
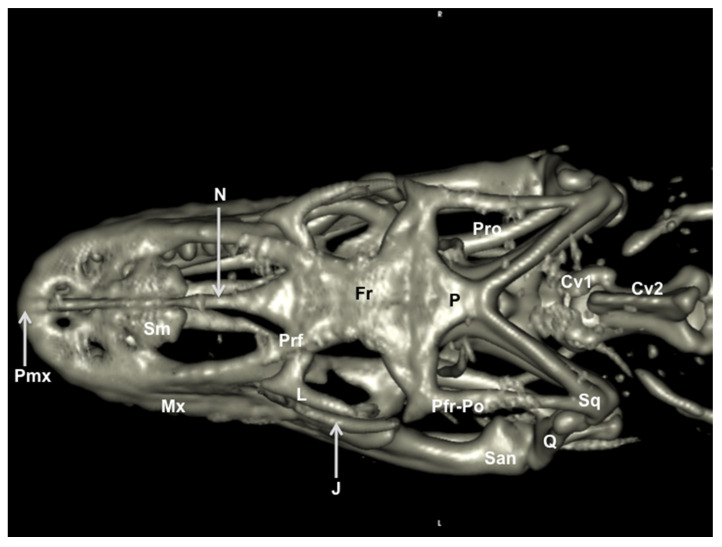
Three-dimensional volume-rendered reconstruction image of the cranium of *Varanus komodoensis*. Dorsal aspect. Pmx: Premaxillary bone. Mx: Maxillary bone. Sm: Septomaxilla. N: Nasal bone. Prf: Prefrontal bone. Fr: Frontal bone. L: Lacrimal bone. J: Jugal bone. Q: Quadrate bone. Sq: Squamosal. Pfr-Po: Postfrontal + postorbital. P: Parietal. Pro: Prootic. San: Surangular bone. Cv1: First cervical vertebra. Cv2: Second cervical vertebra.

**Figure 8 animals-11-01078-f008:**
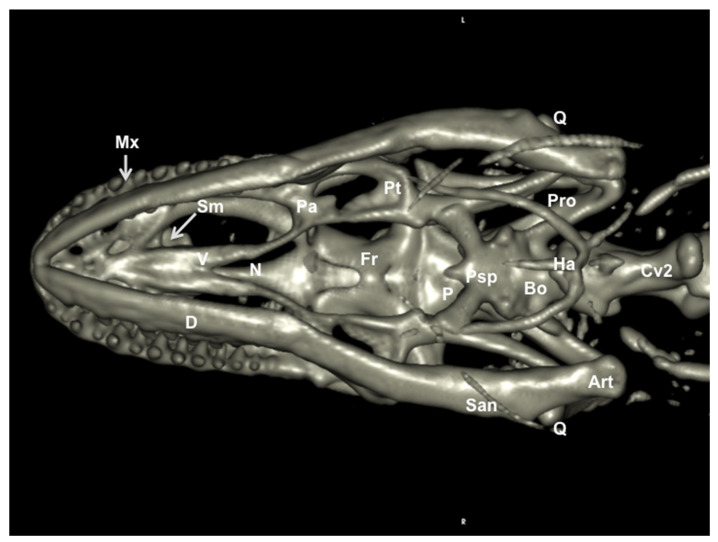
Three-dimensional volume-rendered reconstruction image of the cranium of *Varanus komodoensis*. Ventral aspect. Mx: Maxillary bone. D: Dentary bone. San: Surangular bone. Art: Articular bone. Q: Quadrate bone. Bo. Basioccipital bone. Psp: Parabasisphenoid bone. Pro: Prootic bone. Pt: Pterygoid bone. Pa: Palatine bone. V: Vomer. Sm: Septomaxilla. N: Nasal bone. Fr: Frontal bone. P: Parietal bone. Ha: Hyoid apparatus. Cv2: Second cervical vertebra.

**Figure 9 animals-11-01078-f009:**
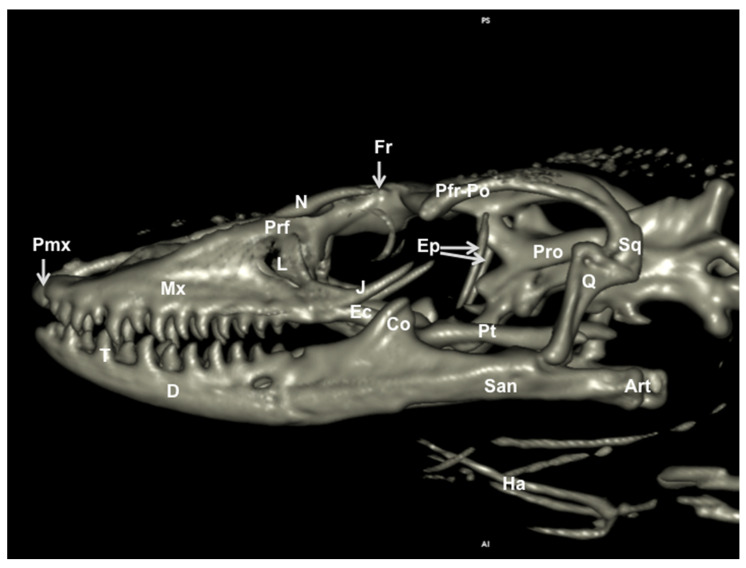
Three-dimensional volume-rendered reconstruction image of the cranium of *Varanus komodoensis*. Lateral aspect. Pmx: Premaxillary bone. Mx: Maxillary bone. Prf: Prefrontal bone. N: Nasal bone. Fr: Frontal bone. L: Lacrimal bone. J: Jugal bone. Ec: Ectopterygoid bone. Pt: Pterygoid bone. Ep: Epipterygoid. Q: Quadrate bone. Sq: Squamosal. Pfr-Po: Postfrontal + postorbital. Pro: Prootic. D: Dentary bone. T: Tooth. San: Surangular bone. Art: Articular bone. Co: Coronoid bone. Ha: Hyoid apparatus.

**Figure 10 animals-11-01078-f010:**
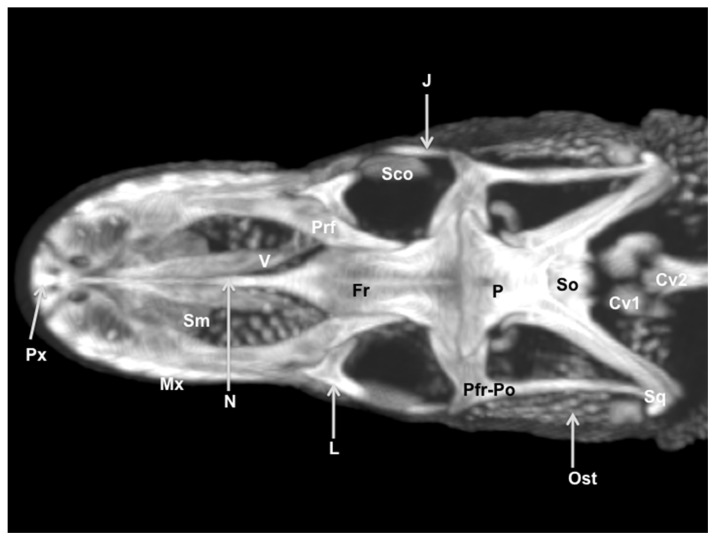
Dorsal MIP image of the cranium of *Varanus komodoensis*. Px: Premaxillary bone. Mx: Maxillary bone. Sm: Septomaxilla. V: Vomer. N: Nasal bone. Prf: Prefrontal bone. Fr: Frontal bone. L: Lacrimal bone. Sco: Scleral ossicles. J: Jugal bone. Pfr-Po: Postfrontal + postorbital. P: Parietal. So: Supraoccipital. Sq: Squamosal. Ost: Osteoderms. Cv1: First cervical vertebra. Cv2: Second cervical vertebra.

**Figure 11 animals-11-01078-f011:**
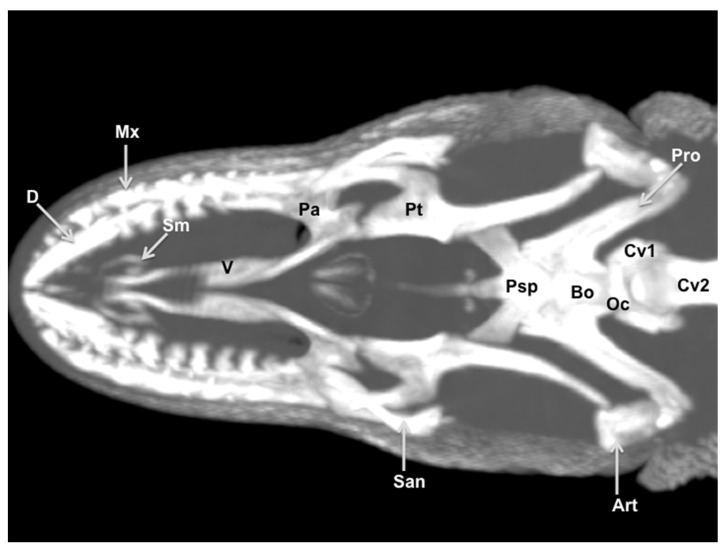
Ventral MIP image of the cranium of *Varanus komodoensis*. D: Dentary bone. Mx: Maxillary bone. San: Surangular bone. Art: Articular bone. Bo. Basioccipital bone. Psp: Parabasisphenoid bone. Oc: Occipital condyle. Pro: Prootic bone. Sm: Septomaxilla. V: Vomer. Pa: Palatine bone. Pt: Pterygoid bone. Cv1: First cervical vertebra. Cv2: Second cervical vertebra.

## Data Availability

Not applicable.

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
