# Peer review of "Cranial Structure of *Varanus komodoensis* as Revealed by Computed-Tomographic Imaging"

_animals, 2021, doi:10.3390/ani11041078_

Round 1
Reviewer 1 Report
Review of Sara Perez et al. MS Animals-1147300, “Computed tomographic anatomy of the head of the Komodo dragon (Varanus komodoensis)”
This is a basic, straightforward and interesting manuscript about CT scans of Komodo dragon heads. It could be a useful addition to the published literature.
The figures are mostly clear and well labeled. Without doubt these are the highlight of the manuscript and the “meat” of the paper. They are of good variety and quality.
The accompanying text is brief, which is fine, although several of the word choices are odd, perhaps because none of the authors are native writers of English. This is fine, but they might wish to rethink some of the words they use.
The authors explain, in the Introduction, that 1) exotic reptiles are becoming increasingly common as pets, and 2) CT scanning is becoming an increasingly common and important diagnostic technique to assess the health of different animals. I agree that both of these statements are true. However, no one has (or is soon likely to have) a Komodo dragon as a pet, not only because of the extreme size and danger posed by this species, but also because (as the authors of this manuscript point out) this species is threatened.
I still think that better understanding the skeletal and other internal structures of Komodo dragons would prove useful, not merely to zoologists and anatomists but also to veterinarians and pet owners. However, this leaves the very important question of how the anatomy of Varanus komodoensis differs from that of other varanid lizards as well as other reptiles. This is a big question that is never addressed here. If the authors cannot address or answer this question, they must at least acknowledge this fact. It is nice to have CT (and related) images of Komodo dragons, but we never learn from this manuscript if Komodo dragons are just really large varanids or if their anatomy differs from that of other varanid lizards in any way, minor or major.
The authors explain why they have no radiographs for comparison, but it would also be nice to have CT scans of dead, skinned animals without the osteoderms that make these images less crisp and useful than they might be.
The skeletal and soft tissue anatomy described here (both in the text and in figure captions) seems quite good, and I find nothing to correct.
The sentence that begins on Line 250 makes it seem as if cited source #11 refers to this species (i.e., V. komodoensis) or groups of Komodo dragons, when it does not. I suggest that the end of this sentence be altered to read “perhaps due to the enormous disparity in the form that evolved among varanid lizards” (or something like that).
Minor points:
Lines 20 and 27: The word “Besides” is used in both the Abstract and Simple Summary (I think in the sense of “also”) but it is unnecessary and can be deleted in both places.
Line 42: I am not sure what the authors mean by “univocal,” a word that I have never seen in English. Unequivocal? I think the authors mean “unambiguous,” and I recommend they use that word instead.
Perhaps don’t begin the paper with the same “In recent years” for the first two paragraphs?
Please spell out what CT means (computed tomography) before this abbreviation is first used (line 48?) in the text. Yes, I know the full term is given in the title and Simple Summary.
Line 69: The dragons’ owner? Or (if different owners for each animal) The dragons’ owners…
It is impossible to see the label for #18 in Figure 2. It has been cropped out of the picture.
What is a maximum intensity projection image? This is not a common technique. Please explain with a brief description.
Line 257: the word “The” does not need to be capitalized in the middle of this sentence.
Line 264: please make clear that the osteoderms are the bones; perhaps rewrite as “…due to bones embedded within the skin. These bones, called osteoderms, vary in shape and complexity.”
Line 272: the first “of” is not needed (“used to diagnose disorders…”)
Line 282: cession?
Line 311: Please put the species name Varanus exanthematicus in italics.
Many of the listed references include other species names that should be placed in italics.
Reviewer 2 Report
Computed tomographic anatomy of the head of the Komodo dragon (Varanus komodoensis)
I was happy to see a manuscript that seemed geared towards a veterinarian audience. The bandwidth of “exotic pets” and captive-held breeding populations is ever increasing, and it is therefore important to establish a basic understanding of the specific anatomy of these animals. In its current form the manuscript appears unfocused, and seems to describe a plethora of findings in details ranging from superficial to precise, which makes it difficult to evaluate its usefulness for anyone. The authors need to state clearly the goal of their study, and then lay out Methods and Results according to that goal. The data presented is interesting, and worthy of publication, but it needs to be formatted to target a specific audience.
There are currently no goals or hypotheses associated with the introduction, and the reader is left to guess what the authors were pursuing by generating and analysing this data. From the images presented I gather that they wished to investigate to what extent structures of the varanid head could be visualized and identified in low-resolution clinical CT-image data. They should address this, and from this motivation generate a few well-defined hypotheses. For example: “All bones of the cranium will be clearly delineated in the CT-image data. Soft-tissue structures will be visible in the image data, but will lack delineation to neighbouring soft-tissue structures.” These hypotheses must then be addressed, in turn, in the Discussion. The latter is currently way too superficial to allow for clinical use of the data provided.
I commend the authors for their efforts in writing this manuscript in English. The language barrier is obvious, but most of the text is sufficiently articulate to communicate the findings described. However, I must point out that the phrasing is often obscure, and sometimes equivocal. For final publication I recommend hiring either an editor, or adding a coauthor with a more solid grasp of the English language.
I further request the acquisition and citation of additional literature. The primary literature harbors a markedly greater variety of information on varanid heads than the authors currently include. Below I quote a few that are of exceptional importance. Robert Mertens worked closely with several Zoos, and published considerable insights into captive-held varanids, particularly regarding their head anatomy. Susan Evans wrote the definitive guide to squamate cranial anatomy.
Overall, this manuscript requires additional work, reconfiguration, and most of all focus towards a clearly defined question that the authors deem important to address. I hope that they authors find this advice useful, and consider resubmission after a thorough overhaul of this study, which may make this an important reference for clinical studies of varanics. Below I provide additional feedback on specific passages of the text.
Mertens, R. (1942): Die Familie der Warane (Varanidae). 2, Der Schädel. Senck. Naturf. Ges., 465:1–118.
Mertens, R. (1948): Über den Komodo-Waran des Berliner “Aquariums,” besonders seinen Schädel. Senckenbergiana 28:153–157.
Evans, S. (2008): The skull of lizards and Tuatara. Biology of the Reptilia, 20:1-347.
1: There is no such thing as “tomographic anatomy”. Do you mean “Investigations of the anatomy using computed tomography”?
Alternative title: “Cranial structure of Varanus komodoensis as revealed by computed-tomographic imaging.”
2: Eliminate the first “and” in your authors list.
17: The summary is too superficial. What it should contain instead: “We investigated the head of Komodo dragons using CT imaging. Cross sections show that all cranial bones can be clearly delineated, while soft tissue structures are evident, but not clearly identifiable without anatomical atlas.”
37: The main body of the text is ripe with grammatical problems, which I won’t comment on, because the entire manuscript needs to be reworked anyway.
42: You make improper use of the word “univocal”. “Solid”, or “general” would be better terms in this context.
57: Why is this important? Varanids in general, and Komodo dragons in particular, are not widely held in private households. I think the motivation behind a clinical study is rather based upon animals held in zoos, rather than treating lizards in the wild, or in public veterinary clinics. You need to focus on one particular audience, and ask yourself what questions that audience might have about CT-visualisation of varanids.
59: While there may be little radiographic imagery available for the cranium of Varanus, other squamates feature quite prominently in radiographic images, providing crucial comparators for your study. You also need to point out why you focus on Komodo dragons. These are the largest extant terrestrial lizards, and such extremes rarely make great comparators for clinical studies. Explain why you picked them, and what the caveats of such a comparison might be.
69: Where were these animals sourced and housed? It is crucial to know how old these animals are, and whether they were held in captivity. Also: how large are the specimens?
77: Is the visual resolution and voxel size of your scan 1 mm? Or is this just an arbitrary threshold that you picked?
79: The term “dragons” is colloquial, and its abundant use therefore inappropriate for scientific literature. Try to avoid its use, and predominantly refer to “specimens” when you address the two specimens that you examined.
80: The term “windows” stems from clinical nomenclature and is, therefore, difficult to understand by non-clinicians, or clinicians who don’t work directly with CT images. It is worthwhile adding a sentence to explain how you are manipulating the histogram of the grayscale distribution, and what those two settings would normally highlight.
85: What is the mechanism behind generating “maximum intensity projection”, and what makes them valuable for this study? What program and settings did you use to acquire MIP images?
86: Bone has a relatively high attenuation relative to most other tissues of the squamate head. MIP is unnecessary to visualize bone in CT images.
89: “transverse sections are provided that demonstrate critical anatomical features of the varanid cranium.”
105: the material filling the medullary cavity should be very low in relative density, and therefore show a low, and not an “intermediate” attenuation. If your findings contradict this, you must explain why.
114: “Sagittal image of the head of Varanus comodoensis.”
116: Many authors seem to deem this information obvious, and therefore leave it out. Thank You for providing information on orientation of the images. It really is crucial.
165: This is not a sentence. Do you mean: “we provide images of the three-dimensional structure of the cranium in dorsal and ventral view (Figs. 7, 8, respectively)”?
168: “The orbital border is circumscribed by the lacrimal […]”
169: “The jugal is clearly distinguishable from the ectopterygoid.”
172: “In ventral view the following bones of the neurocranium are clearly delineated: parabasisphenoid […]”
205: MIP provides composite images of density contrasts throughout the image stack, and seems to provide a great density contrast between bones. In clinics lacking CT-capabilities, could a 2D radiograph deliver similar results? To what degree could these images be diagnostic?
242: I don’t understand this sentence. You are not “contrasting” the previous statement.
243: That is not true. The selected imaging technique depends foremost on the clinical question. Computed tomography has considerable disadvantages, compared to other imaging techniques, particularly the investment of time and money required to obtain images. You must discuss these differences, and alert the reader to 1) information that can be obtained from 2D radiographs, and 2) information that is only obtainable through CT images.
246: Do you mean “soft-tissue differentiation”? I do not see that evidenced in your images, which show almost exclusively mineralized tissue.
248: I presume that this is the actual intention of your manuscript. If so, you should build your introduction towards this question: “what tissues can be clearly delineated in low-resolution clinical scans of the varanid cranium?”
251: This is not true. The squamate cranium in general, and that of varanids in particular, has featured rarely in morphometric comparisons. Your absence of citations for this sentences proves as much. This also directly contradicts a statement you made on line 59.
257: This sentence indicates that you performed various CT scans of each specimen, which is not true. Instead, you analysed one image stack with different visualization methods. This variation in visualization is the actual strength of your study, and you should highlight it. How much information can a clinician gain from one set of CT data, and how can the images be manipulated to reveal these informations?
262: The primary cause for low-resolution images is a low-resolution scan. In order to address this topic you need to divulge the resolution of your scans, and the size of the specimens. Please add scale bars to all your images.
Your wording indicates that you are concerned about your ability to discern different skeletal structures in your CT image data, which is not just a result of image resolution, but also density contrast. You need to discuss those two issues separately.
264: Osteoderms are clearly visible in all your images, and potentially mask some of the anatomical detail in MIP images. However, osteoderms appear easily distinguishable from bones in your cross sections. Discuss this.
266: It would be valuable to compare images obtained via 2D radiography and MIP mapping of CT data. A clinician must know what the limitations are of either technique, as he/she must then decide whether application of the much more expensive CT-technology is worth the added value of the resulting images. This difference should be the focus of your investigation, and is, sadly, not addressed at all.
270: Beyond the cranium, mandible, eye balls, and airways you do not show much potential for tissue differentiation. It would be better to subdivide your Results section into paragraphs pertaining to the 1) hard tissue, 2) soft tissue and 3) airways, pointing out which anatomical system each of these represents, how they can be visualized, and how pathologies might be recognized.
Figure 1-6: Providing inset A with every figure is unnecessary, as the image is identical in six consecutive figures. Mark the locations of all cross sections on Figure 1A; then delete A from all subsequent images.
Figures 7-9: Not all of the borders you describe in the text are evident to me. It would be useful to provide an inset with each figure, containing a sketch of the cranium with delineations between bones.
Fig. 11: MIP provides a univocal density map, and should therefore provide symmetrical images of dorsal and ventral view, similar to a radiograph. Your dorsal and ventral MIP renderings differ markedly. Why is that? I suspect that you applied different settings for the two images, but you do not address this.
Literature 1: species names must be italicized.
Literature 13: Are you referring to chapter 3.3? It has two authors.
Literature 9: websites need a citation date.
Round 2
Reviewer 2 Report
I am pleased to see the remarkable advances that this manuscript has made in the past few weeks. Content, layout, and phrasing have improved markedly. The outline and purpose of this study are clear; the figures support the discussion and context. The writing is succinct, and outlines all techniques applied to a degree that makes them reproducible in a clinical setting.
As an overly picky reviewer I have a short list of comments and suggestions on several grammatical and contextual issues (see below), but I consider these minor issues, which can be passed without additional review from my side.
The authors have put a lot of thought into the presentation of their findings, and the images presented could well serve as a layout for clinical research and diagnosis, particularly for large-bodied varanids, regarding which detailed anatomical knowledge is still lacking. The authors could take one additional step to increase the applicability of their study. All major structures of the cranium, and gastrointestinal tract are outlined in the images provided, but cross-referencing between these images with only the numbers at hand makes it difficult to employ them as a clinical tool. Findings distinct structures in these images would be markedly easier, if the authors were to replace most of the numbers with abbreviations, indicating a standardized system. Evans (2008) employed a very common standard for the abbreviation of skeletal structures, and using her system would make the vast majority of the visualized anatomical details immediately recognizable throughout all figures.
44: “In recent years the contributions of zoo veterinarians …”
46: “that threaten the survival of species …”
50: “, but only sparse numbers …”
52: Strike “However”, and start that part of your argument in line 50 (see above)
54: ”complex structure, which is challenging …”
55: Strike the “Therefore”
63: I assume you refer to snout-vent length. Be specific.
64: strike “examination”
66: “The owner of the animals” à do not overuse the term “dragon” This is one of two locations in your manuscript where it still seems out-of-place.
84: “… pulmonary window can be applied, delivering alternate streams of information.”
95: Great job on the revised figures!
102: “enabled us to identify …”
102: “These features were identified …”
105: “With regards to hard tissue, …”
107: “Thus, the bones of the cranium …” – ‘skull’ is a term laden with anatomical conflict.
113: “Air-filled structures, such as the nasal cavity, larynx, trachea, and the oral cavity gave negligible CT-tissue density and appeared black with both window settings.”
116: “Soft-tissue structures, such as the jaw muscles”
182: “in lateral and ventral view.”
183: The tooth row curves with the margin of the mandible and maxillary. It is therefore not “straight”. Also point out that the primary curvature of the maxilla is convex, whereas that of the mandible is concave. This is not trivial.
184: “sur-angular” – respect the Latin
205 (and following figure captions): “… of the cranium of Varanus komodoensis.” – you lack the comparative evidence to call it “normal”, and it is easy enough for you to avoid such a discussion. Also, you show the cranium, not the entire head. Adopting the suggested wording makes the captions a bit smoother, and avoids conflict.
213: A voxel size of 1 mm is not ‘excellent resolution’ in absolute terms, but it is formidable for veterinary scans. “These images were able to resolve the relation …”
214: strike the “Therefore,”
215: “We are also able to show how the the laminar disposition of the vomer supports the septomaxilla (Fig. 10).”
222: “… excellent visualization of the pterygoid, a flat, Y-shaped bone.”
222: ” This bone provides a rounded process that contacts the caudal border of the palatine.”
249: “third and fourth …”
250: Fast imaging is an advantage of radiography, not of CT-scans. One image is taken faster than thousands.
255: “The head of the Komodo dragon …”
257: “The head of this iconic varanid represents a complex structure, composed of various tissues with varying degrees of attenuation in radiographic images, making it a challenging object to assess.”
262: “Visualising images through use of the “bone window” provided good resolution for skeletal structures, whereas the “soft tissue window” allowed us to distinguish …”
266: What is this “low definition” you are addressing here? Is this a critique on soft tissue resolution in our own study or in other studies? I don’t understand what you mean to address.
274: Strike the first half of the sentence. You obtained radiographic images, and compiled them into a three-dimensional visualization. “Employing computed tomography we were able to fully visualize the cranium in virtual reconstructions and MIP images.”
279: “This morphology contrasts with that of most other varanids, which feature a relatively narrow rostrum, a dorsoventrally tall cranium, and a straight ventral margin of the maxilla.”
281: Are you pointing out variation within lizards, or differences between varanids and other lizards, or variation within varanids? This sentence is unclear to me.
285: “MIP images proved a helpful tool in delineating bones in volume-rendered images.”
287 – 290: These two sentences belong onto the end of your Methods.
292: “Of the head of Varanus …”
Figure 7: the arrow (4) points towards the number; it should point at the structure.
Author Response
Answer to Reviewer: 2
We sincerely appreciate all valuable comments and suggestions, which helped to improve the quality of our manuscript. We have resubmitted our manuscript with all the modifications, which have been highlighted by using Track Changes and are listed below.
- As you recommend, we have replaced the numbers with abbreviations, following the standardized system provided by Evans (2008).
- Line 44: “In recent years the contributions of zoo veterinarians …” this sentence has been modified according to your comment
- 46: “that threaten the survival of species …” this sentence has been modified as follows “that threaten the survival of species in wildlife conservation have increased”
- 50: “, but only sparse numbers …” The modification has been added.
- 52: Strike “However”, and start that part of your argument in line 50. As you suggest, we have deleted this word, and start our argument with “To date”.
- 54: ”complex structure, which is challenging …” The modification has been included.
- 55: Strike the “Therefore”, this word has been deleted.
- 63: I assume you refer to snout-vent length. Be specific. The requested information is included in this section.
- 64: strike “examination”, the word has been deleted.
- 66: “The owner of the animals” à do not overuse the term “dragon” This is one of two locations in your manuscript where it still seems out-of-place. This modification has been included (now line 68)
- 84: “… pulmonary window can be applied, delivering alternate streams of information.” The sentence is included in material and methods section.
- 105: “With regards to hard tissue, …” The sentence has been modified as you recommend
- 107: “Thus, the bones of the cranium …” – ‘skull’ is a term laden with anatomical conflict. The word has been replaced as you recommend
- 113: “Air-filled structures, such as the nasal cavity, larynx, trachea, and the oral cavity gave negligible CT-tissue density and appeared black with both window settings.” The sentence has been modified following your recommendation.
- 116: “Soft-tissue structures, such as the jaw muscles”.The sentence has been modified following your recommendation (now Line 117).
- 182: “in lateral and ventral view.” The word has been included following your recommendation (now Line 187).
- 183: The tooth row curves with the margin of the mandible and maxillary. It is therefore not “straight”. Also point out that the primary curvature of the maxilla is convex, whereas that of the mandible is concave. This is not trivial. The entire sentence has been modified, including your recommendation
- 184: “sur-angular” – respect the Latin. Following the standardized system provided by Evans (2008), surangular was the correct form.
- 205 (and following figure captions): “… of the cranium of Varanus komodoensis.” – you lack the comparative evidence to call it “normal”, and it is easy enough for you to avoid such a discussion. Also, you show the cranium, not the entire head. Adopting the suggested wording makes the captions a bit smoother, and avoids conflict. We have changed the figure captions, deleting “normal” and “head”, this last has been replaced by “cranium”.
- 213: A voxel size of 1 mm is not ‘excellent resolution’ in absolute terms, but it is formidable for veterinary scans. “These images were able to resolve the relation …” The modification has been included (now Line 222)
- 214: strike the “Therefore,” This word has been deleted
- 215: “We are also able to show how the the laminar disposition of the vomer supports the septomaxilla (Fig. 10).” The modification has been included (now Line 224-225)
- 222: “… excellent visualization of the pterygoid, a flat, Y-shaped bone.” The modification has been included (now Line 231)
- 222: ” This bone provides a rounded process that contacts the caudal border of the palatine.” The modification has been included (now Line 232)
- 249: “third and fourth …” The modification has been included (now Line 259)
- 250: Fast imaging is an advantage of radiography, not of CT-scans. One image is taken faster than thousands. This part of the sentence has been deleted.
- 255: “The head of the Komodo dragon …” The modification has been included (now Line 265)
- 257: “The head of this iconic varanid represents a complex structure, composed of various tissues with varying degrees of attenuation in radiographic images, making it a challenging object to assess.” The entire paragraph has been modified as you recommend (Line 267).
- 262: “Visualising images through use of the “bone window” provided good resolution for skeletal structures, whereas the “soft tissue window” allowed us to distinguish …” The paragraph has been modified as you recommend (Line 272).
- 266: What is this “low definition” you are addressing here? Is this a critique on soft tissue resolution in our own study or in other studies? I don’t understand what you mean to address. We are addressing the low soft tissue resolution observed in our study, which was similar to other studies performed in reptiles with similar CT equipment.
- 274: Strike the first half of the sentence. You obtained radiographic images, and compiled them into a three-dimensional visualization. “Employing computed tomography we were able to fully visualize the cranium in virtual reconstructions and MIP images.” The paragraph has been modified as you recommend (now line 284)
- 279: “This morphology contrasts with that of most other varanids, which feature a relatively narrow rostrum, a dorsoventrally tall cranium, and a straight ventral margin of the maxilla.” The paragraph has been modified as you recommend
- 281: Are you pointing out variation within lizards, or differences between varanids and other lizards, or variation within varanids? This sentence is unclear to me. The sentence was clarified, it was a difference between varanids and other lizards.
- 285: “MIP images proved a helpful tool in delineating bones in volume-rendered images.” The entire sentence has been modified as you recommend
- 287 – 290: These two sentences belong onto the end of your Methods. We have moved these sentences to material and methods section.
- 292: “Of the head of Varanus …the sentence has been corrected
- Figure 7: the arrow (4) points towards the number; it should point at the structure. As you suggest, we have corrected the arrow, now it points at the structure.